# Epidermal necrolysis (Stevens-Johnson syndrome/ toxic epidermal necrolysis) as extensive boundary violation: A qualitative study on the illness experience and care needs of survivors in the context of the first German guideline

Ruben Heuer[1,*,‡], Maren Paulmann[2‡], Maja Mockenhaupt[2], Alexander Nast[1]

1 Division of Evidence-based Medicine (dEBM), Department of Dermatology, Venereology und Allergy, Charité - University Medicine Berlin, Corporate Member of Free University of Berlin, Humboldt University of Berlin, and Berlin Institute of Health, Berlin, Germany, 2 Dokumentationszentrum schwerer Hautreaktionen (dZh), Department of Dermatology Medical Center, University of Freiburg, Freiburg, Germany

‡ These authors share first authorship on this work.
* ruben.heuer@charite.de

## Abstract

### Background

In response to considerable heterogeneity in German healthcare for patients with epidermal necrolysis (EN; also Stevens-Johnson syndrome/ toxic epidermal necrolysis), a national guideline was developed. However, no patient initiatives were identified to represent patient preferences, which have yet to be systematically evaluated in Germany.

### Objective

We conducted a qualitative study on EN survivors' experiences to collect contextualised data on patient preferences and identify potential shortcomings and untapped potentials in routine care.

### Methods

We contacted 14 participants who were either survivors of EN or family members of survivors. After recruitment and obtaining informed consent, we conducted individual and dyadic interviews using a semi-structured guide developed in a preceding focus group with the same patient collective. Following grounded theory principles, recruitment was contingent on emerging themes. In our analysis, we coded interview data into thematic categories of increasing abstraction, resulting in a conceptual framework accounting for universal features of the illness experience.

**Data availability statement:** Anonymized transcription excerpts without indirect identifiers are provided in the manuscript and the supporting information file. Due to the very low prevalence of the disease under investigation, data was collected from a small group of participants. These data, containing indirect identifiers, are available from the Division of Evidence-Based Medicine (dEBM) at Charité Universitätsmedizin Berlin (contact via debm01@charite.de) for researchers who meet the criteria for access to confidential data.

**Funding:** This research was funded through the federal joint committee of the joint self-government of physicians, dentists, hospitals, and health insurance funds in Germany (https://www.g-ba.de/; grant number 01VSF21002). The funders had no role in study design, data collection and analysis, decision to publish, or preparation of the manuscript.

**Competing interests:** Conflict of Interest Disclosures: Maja Mockenhaupt reported serving on a scientific advisory board for Boehringer Ingelheim and Merck AG; the German Registry on severe skin reactions (dZh) received financial support for research (through contracts with the University of Freiburg) from several pharmaceutical companies for the provision of safety information related to drugs manufactured or distributed by them. Ruben Heuer, Maren Paulmann, and Alexander Nast declare no potential conflicts of interest involving the work under consideration for publication. This does not alter our adherence to PLOS ONE policies on sharing data and materials.

## Results

From survivors' perspectives, EN represents a profound transgression or destabilisation of personal and interpersonal boundaries, affecting several fundamental areas of human experience. Even after returning to domestic life, survivors report physical limitations that, despite their perceived invisibility to their social environment, lead to permanent changes in self-concept and personal values. Conversely, the majority of participants emphasised restoring lost boundaries in leading an authentic and meaningful life and countering social alienation. The study identifies five key dimensions of boundary violations caused by EN and four areas where healthcare providers can help restore boundaries to facilitate successful coping.

## Conclusion

Healthcare providers should be mindful of life-altering implications of EN and how debilitating illness sequelae can affect survivors beyond a transient emergency. Optimal medical care should facilitate stabilising lost boundaries over the entire recovery period and requires a sensitivity to existential aspects of the disease only accessible through a sustained dialog with patients and family members.

## Introduction

One of the most severe dermatological emergencies, the adverse skin reaction epidermal necrolysis (EN; also Stevens-Johnson syndrome/ toxic epidermal necrolysis) often results in traumatising periods of life-threatening illness. In contrast to burn injuries, EN presents as a progressive disease that can leave patients overwhelmed with rapid deterioration [1–3]. Different from burn survivors, many EN-patients also experience extensive mucosal damage, leading to complications of the oral cavity, eyes, and genitals. Among those, conjunctival or corneal epithelial sloughing requiring rigorous ocular surface protection, and painful oral erosions impairing oral nurtition are some of the most debilitating [1–3]. Long-term sequelae that demand continued specialist care are also common, including ocular surface scarring, vulvovaginal pain or phimosis [4–8]. Moreover, survivors can suffer psychological consequences of EN, as anxiety around recurrence and avoidance of potentially causative medications can create tremendous emotional strain [4,6–13].

Experiencing skin failure on a large scale, losing one's most intimate source of protection against the outside world can mean enduring a period of extreme vulnerability that is felt years after hospital discharge [4–14]. Qualitative studies in EN survivors consequently highlight the importance of clear and reassuring communication from medical providers in the early treatment phase. Yet, this demand is difficult to meet for personnel that mostly lacks disease-specific experience due to the very low incidence of the disease [13,15–17]. It is therefore not surprising that some survivors lose confidence in the medical system [16,17]. This compromised trust may lead to harmful underutilisation of healthcare resources and thereby perpetuate the influence

of the disease on the day-to-day lives of the affected [18–20]. But even in those struck most severely, narratives of lasting personal growth attest to the existence of a new normal. Accordingly, studies report long-term reductions in quality of life in EN survivors, while some maintain or even exceed their self-reported levels of psychosocial wellbeing [4,9–13]. These findings raise several questions that extend beyond the purview of quantitative inquiry. Furthermore, they also remain largely unaddressed by the extant qualitative literature, which has predominantly focused on the relationship between patients and healthcare providers in acute care settings and less on the determinants of successful long-term rehabilitation. What distinguishes those recovering physically and emotionally from those who only accomplish the former? How do patients conceptualise their condition, and what steps do they take to rebuild their lives? How do they navigate the period from a state of powerlessness back to a renewed sense of agency? What information do they receive, and what do they wish they would have been told?

In a two-year period from July 2022 to July 2024, an interdisciplinary expert committee developed the first national guideline in response to a pronounced heterogeneity in medical care for EN in Germany [21,22]. The purpose of this study was to aid guideline developers in considering the experience of survivors and their family members as they pertain to the questions outlined above and to systematically capture unmet needs, build patient representation, and improve the management of this life-altering condition.

## Methods

Before data collection, we obtained approval from the ethics committee of Charité – Universitätsmedizin Berlin (EA2/096/22) and written consent from each participant, or, in the case of adolescent participants, their parents or legal guardians. Confidentiality and anonymity were ensured by pseudonymising transcripts and following a pre-specified data protection protocol. Between August 4 and November 29, 2022, we conducted and analysed 12 interview sessions, including nine one-on-one interviews, two dyadic interviews, and one focus group conversation, recruiting from the German Registry of Severe Skin Reactions (Dokumentationszentrum schwerer Hautreaktionen; dZh). We included survivors aged 16 and over with a confirmed diagnosis of EN, and, where applicable, their relatives. We excluded individuals with significant cognitive impairments that could compromise their capacity to provide informed consent.

Aiming to moderate the impact of the researchers' preconceptions in selecting patient-relevant questions, we co-created a loosely structured interview guide in an online focus group session with three survivors and one family member. As ultimately no methodological decision can nor should fully remove the researchers' influence from the results of qualitative inquiry, a reflexive stance was adopted throughout the study [23–25]. During the session, participants, selected to represent different stages of recovery, were asked to modify or add to preliminary questions provided by one of the study authors (RH) to elicit personal reflections. The finalised guide was then used as a reference during further data collection and comprised questions such as "if you/your relative(s) felt unsettled by the skin reaction, what measures helped you to gain a better understanding of your situation?". In addition to increasing the interview guide's validity, we chose the focus group setting to integrate personal interpretations of the illness experience with those of other participants at a higher conceptual level, fostering collective meaning-making. Paradigmatic ideas and interpretive patterns used to characterise episodes of EN, so-called sensitising concepts, informed the consecutive interviews by providing a lens through which to encounter novel data without imposing a preconceived epistemological framework [26–28].

For our study, we adopted an inductive, grounded theory-based approach to work from the individual experiences of survivors and their family members toward a theoretical account of illness-specific care needs [28–30]. As qualitative researchers, we assumed that medical concepts neither exhaustively describe nor explain reality. Our analysis rather focused on the interpretive processes of recounting, sharing, and reframing of a disruptive life event by which multiple potential meanings are channelled into a coherent biographical narrative [31,32]. In line with an interactionist perspective, the interviewer (RH), a social scientist with a background in physical therapy, did not present in conversation as a blank slate but shared personal experiences relating to the patients' reports when appropriate [33,34]. This approach was

chosen to promote a spontaneous narration of survivors' life stories as they relate to those of others, touching on matters of personal identity and the disease's existential import.

The interviews with nine EN survivors and five family members were performed between August and October 2022. All interviews took place online using the platform Skype for Business with the exception of two, one conducted at the patient's bedside during hospital treatment for EN and one at the researcher's office (for additional information, see Table 1). No time limit was set for the interviews, which lasted between 26–130 minutes for a mean duration of 76 minutes across sessions. Additional information regarding recording and software used can be found in the supplement (S1 File). Sensitising concepts from the preparatory focus group discussion grounding the main analysis as well as additional interview quotes are likewise presented in S1 File.

Consistent with the constant comparative method of grounded theory, sampling and analysis were conducted concurrently, with each task shaping the trajectory of the other [28]. Within the limits imposed by the rarity of the disease, we tried to purposefully sample new participants based on emerging insights, such that gaps in the data could be closed by subsequent interviews. Coding and analysis followed a course of incremental abstraction, moving from line-by-line coding to creating larger conceptual themes. This process was guided by written memos to explore connections between themes, reflect on the researchers' role in code development, and record tentative hypotheses, resulting in a theoretical framework to account for shared features of the participants' narratives. Data collection was discontinued when theoretical saturation was achieved, defined as the point at which no new information emerged that was unaccounted for by the developed framework [35]. To validate the insights gained in the process, one participant was invited for a second interview at the end of data collection to discuss the preliminary theoretical framework. As an additional form of member checking in the context of the larger guideline project, we enlisted two participants in the guideline committee, the group tasked with authoring the manuscript and voting on the included recommendations. Furthermore, participants were consulted on issues specific to the medical treatment received (e.g., in-patient follow-up care, which is currently not standardised for EN) or assisted in drafting and reviewing chapters informed by the study results.

## Results

### Themes

Universally, EN survivors perceived their skin reaction as a life-altering event. Most found themselves profoundly destabilised in terms of identity, long-held beliefs about illness and health, as well as social relationships, leading to a re-evaluation of core values better fit to support bodily self-determination. EN figured in the lives of survivors as an intimate encounter with personal boundaries, here understood as the perceived limits of the self that relate to identity, integrity, and safety from outside influences. In doing so, they went from periods of violent transgression or breakdown to asserting newfound limits. Five themes representing personal boundaries touched on by the EN experience emerged from the analysis: stress limits/breaking points, boundaries between wake/sleep or fantasy/reality, limits of understanding, social

**Table 1. Recovery periods at the time of interview, sex, and age of participants**
**(italic = female, non-italic = male; bold = survivor, non-bold = family member).**

|  | Age: < 25 | Age: 25–59 | Age: ≥ 60 |
|---|---|---|---|
| ≤2 years after reaction | *DZH001a*<br>**DZH002** | ***DZH005*** **KFA002** *KFA002a*<br>*DZH001*<br>DZH005a<br>*DZH006* | ***CCM001***<br>**DZH004** |
| 3-10 years after reaction | **DZH003a** | ***DZH007***<br>*DZH003* | – |
| >10 years after reaction | – | **KFA001** | – |

boundaries, and temporal boundaries (see Table 2). For a schematic representation of the different dimensions of boundary violations across time, see Fig 1.

## Stress limits/breaking points

Notably, survivors and their family members *felt overwhelmed* at different times in the recovery process. While survivors often did not perceive the acute threat to their survival directly due to general anaesthesia or delirium, resulting in limited

Table 2. Themes and subthemes.

| Themes | Subthemes | Summary |
|---|---|---|
| Stress limits/breaking points | Feeling overwhelmed<br>Loss of agency<br>Struggling to adapt | Participants face challenges perceived to lie beyond what they can bear. Confrontation with a potential point of collapse can be traumatic but may also bring about a newfound sense of resiliency. |
| Boundaries between wake/sleep or fantasy/reality | Dream-like experience<br>Normality<br>Hope and Confidence | The sudden rupture in the participants' daily lives in combination with physical overwhelm and an unfamiliar environment creates a sense of unreality. |
| Limits of understanding | Reduced situational understanding<br>Existential distress<br>Understanding of others | Struggling to make sense of their situation, participants can feel misinformed by medical staff. Outside of the hospital, they encounter a lack of understanding for the significance of their health crisis in friends and family as well as in healthcare providers. |
| Social boundaries | Isolation<br>Value shifts<br>Self-determination | During hospitalisation, EN-patients are displaced from their familiar social circles. After discharge, they can experience alienation from their peers due to profound value shifts inspired by critical illness. |
| Temporal boundaries. | Patience<br>Discontinuity<br>Forward-looking perspective | The participants' sense of time is disrupted in multiple ways compounding the incisive character of the illness experience. |

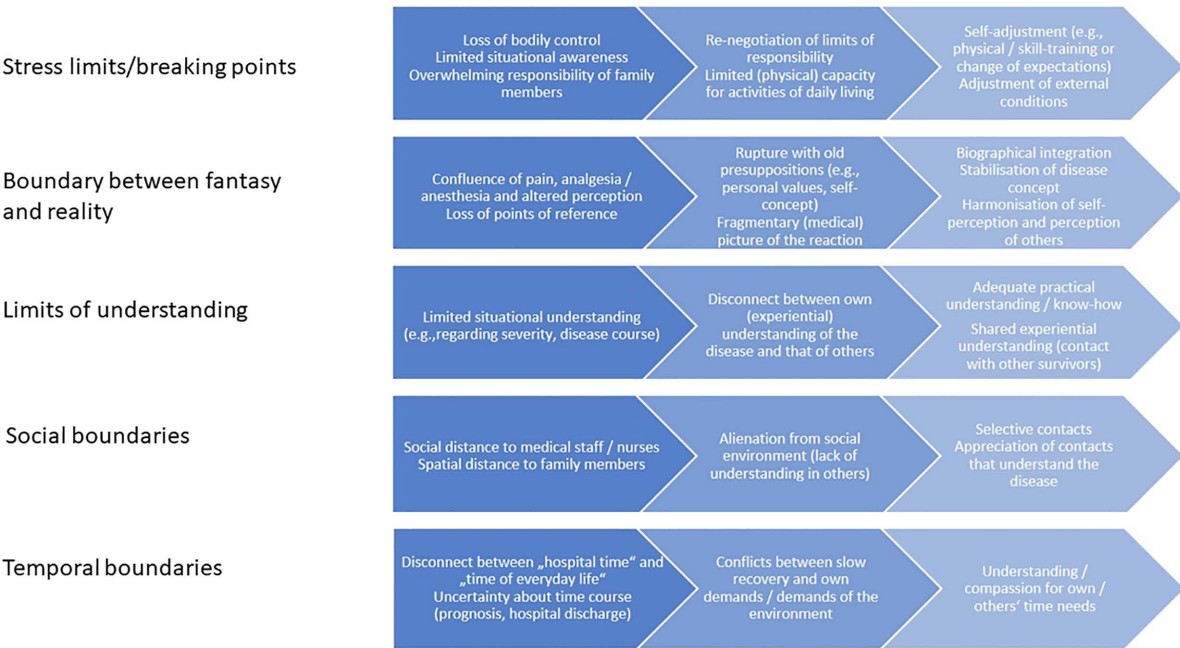

**Fig 1. Dimensions of EN-induced boundary violations across time.**

situational awareness, their family members felt both the anxiety associated with critical illness and the weight of their responsibility as caregivers with full force.

A *loss of agency* and bodily control was among the most prevalent contributors to reaching one's psychological stress limits. Frequently being fully immobilised, survivors rested their gaze on the same spot for hours at a time in case their eyes had not already been dressed shut to prevent further damage. The near-complete loss of agency over their sensory input was often accompanied by intense pain and bouts of insomnia that were not always managed as desired. Despite occasional reports of transparent and empathic communication, survivors and family members sometimes did not understand the reasons for a given treatment decision and felt they had little influence over it.

For all patients, restricted agency and a limited physical capacity for activities of daily life persisted after hospital discharge and were initially met with a *struggle to adapt.* Transient illness sequelae such as nail loss, skin dyspigmentation, sensitivities to certain foods as well as more debilitating, permanent consequences, such as chronic eye inflammation, came unannounced and required extensive adaptions. In the extended recovery period, survivors underwent a process of re-negotiation around the limits of responsibility, reflecting on their own authority over their health, the involvement of the medical system, and the support of their caregivers.

In contrast to the powerlessness felt during hospitalisation, most survivors went on to reclaim a type of agency that respected their newfound boundaries. These boundaries now formed an integral part of their identity, as they adjusted their expectations and behaviour (e.g., acquiring new health skills) as well as their external conditions to suit their needs.

For representative quotes related to the theme outlined above, see Table 3.

## Boundaries between wake/sleep or fantasy/reality

For some patients, a confluence of severe pain, the side effects of analgesia, residual shock, and monotonous sensory input created a sense of unreality. This tenuous grasp on the external world compounded feelings of cognitive destabilisation and displacement from their familial environments or other points of reference and was often described as a *dream-like experience*. Interestingly, similar language was used by family members, who developed a narrow focus on the immediate care needs of their loved ones, often to the exclusion of the external demands of their previous lives, creating a state of uprootedness.

For survivors, stabilising the hazy boundary between fantasy and reality was associated with establishing control over one's perceptual input. As freedom of movement increased, it meant creating a sense of *normality* and continuity with the outside world. In this phase, everyday activities such as listening to radio plays, reading letters from home, and having a snack at the cafeteria presented meaningful steps toward a more hopeful future.

Repeatedly, the path back to reality was recounted as one of regaining *hope and confidence* while old presuppositions regarding personal values or a self-concept incompatible with the lived illness experiences made way for new ones. This

**Table 3. Interview data demonstrating themes related to stress limits/breaking points.**

| |
|---|
| **Feeling overwhelmed** |
| *DZH006 (family member): So, he actually had a bit of a slump every now and then, where he said 'and now it's over, and now we'll do away with it all and then we'll go home'... So that... it was just too much for him. […] Yes, he just didn't feel up to it anymore. He didn't want to go on anymore. He also said a few times 'and then I'd rather die before I... have to lie here any longer'.* |
| **Loss of agency** |
| *DZH001a (survivor): I couldn't sleep and had a panic attack at night – then I pressed the button, then a nurse came, and the nurse called the doctor. And she said, yeah, if you can't sleep – because I asked if I could have a sleeping pill – she said, no, but if you can't sleep, then just read a book.* |
| **Struggling to adapt** |
| *DZH003a (survivor): And in general, it's just a much bigger burden because you… actually at the time when the blisters get bigger, you can no longer go to the toilet on your own and so on or even walk. I'd say it's just like that you feel like an old grandma who's now a care case. But I was fourteen and not really a care case. [...] It's extremely embarrassing when you have to ask someone if they can help you to go to the toilet. Yes, I would say that's just the first…. Yes, it was the first time I was really at my mental limit (long inhalation).* |

process involved biographical integration, where survivors gradually incorporated the illness experience into their personal life stories. At the same time, the survivors' initially fragmentary understanding of relevant medical facts became increasingly sophisticated, suggesting a stabilisation of the disease concept as they developed a clearer grasp of the physiological processes underlying their health condition.

Reengaging with their social environments, survivors also experienced a harmonisation of self-perception and the perception of others. They found ways to convey their illness's significance to family and friends while adjusting their self-image in the face of social reminders of otherness. In some cases, this process was punctuated by moments of despair that lessened only with the disappearance of EN's traces on the body months or years into rehabilitation. Others maintained a positive outlook from the outset. Upon questioning, those less troubled by their skin reaction stated trust in their medical care, in one case also supported by religious beliefs, as a reason.

For representative quotes related to the theme outlined above, see Table 4.

## Limits of understanding

A recurring theme that further added to the surreal quality of the experience was a *reduced situational understanding* in survivors caused by the lack of information on prescribed interventions and the predicted time course of their hospital stay. Mainly, this involved being unaware of the life-threatening character of the disease and not understanding why a particular measure had been taken, in one case to the point of unknowingly waking up catheterised and fully dressed in bandages. Prompted to elaborate by the interviewer, the participants viewed the lack of information on severity and prognosis with ambivalence. While it was universally acknowledged that the impact of the disease came as a shock, most participants expressed concerns they would have lost hope in their eventual recovery had they been more truthfully informed. In contrast, survivors and family members presented unequivocally critically towards the lack of education on potential illness sequelae and, in some cases, held the suspicion that future health problems could have been prevented with better care. A similar contention was expressed by survivors who received conflicting information from different staff members or felt misled when their recovery did not meet the timeframe of all too optimistic predictions.

Especially young survivors, experienced *existential distress,* culminating in the question of why it was them that had been struck by the disease.

Study participants also confronted *limits of understanding in others* unfamiliar with the skin reaction, who frequently did not recognise its significance to the survivors' lives and the impairments it brought about. A perceived disconnect between their own understanding of the disease and that of others led most participants to believe that only going through the experience oneself permitted true understanding and expressed the wish to connect with other survivors.

Additional difficulties in finding understanding pertained to healthcare professionals consulted by survivors after hospital discharge. The specificity of EN-aftercare called for specialised professionals who were scarce and often necessitated extensive travel. Furthermore, as some were still doubtful about the safety of new medications, physicians unfamiliar with

**Table 4. Interview data demonstrating themes related to the boundary between fantasy and reality.**

| |
| --- |
| **Dream-like experience** |
| *DZH003a (survivor): It's quite strange… just so surreal, somehow… I couldn't believe it at all what happened. Everything was normal, and suddenly this hard event comes and kicks you completely off track. That was also...somehow...the dreams, they were just totally real at that time, and it was also difficult to distinguish dream from reality, because just no new input came...* |
| **Normality** |
| *DZH001a (survivor): "I wouldn't have needed the [talk] therapy, in terms of someone listening to me so I can talk about it – that wasn't really the point. I could talk about it all day anyway. But just someone showing me – not just telling me, but actually showing me – that it can become normal. I'm not so... lost now.* |
| **Hope and Confidence** |
| *DZH001a (survivor): Yes, exactly. And that's just... as soon as you're told that your skin has been removed and as soon as you're lying there with the bandages, I'd say that's just... From then on, I would say I had no hope at all. I only saw hope again when the bandage was gone altogether...* |

the disease were seen as incapable of assessing the risk of recurrence. Finding trustworthy physicians was accordingly depicted as a time-consuming enterprise. However, many patients and parents used the opportunity to contact the dZh in terms of questions or problems and stated that they received timely answers and individual advice.

Some participants in the later stages of recovery emphasized the importance of practical know-how regarding activities of daily living. Acquiring an adequate understanding of their impairment was supported by interactions with other survivors, where they exchanged practical knowledge grounded in shared experiences.

For representative quotes related to the theme outlined above, see Table 5.

## Social boundaries

During the acute phase, survivors reported intense feelings of *isolation* amplified by the monotonous sensory profile of their bedridden days and the spatial distance to their family members. During the extended process of recovery, especially patients who experienced their skin reaction early in life witnessed far-reaching changes to their social environment, partly due to a lack of understanding in others, partly as a consequence of growing experiential distance during the fast-moving time of adolescence. Feelings of alienation and shame for visible traces of the disease added to the burden of readjustment. In these cases, successful reintegration was conceived as a process of reclaiming control over one's body, for example, through intense exercise and a process of building new, more selective contacts while fostering existing relationships based on shared values.

Regarding these values, most survivors underwent significant changes prompted by their skin reaction that were generally perceived as hallmarks of personal growth. Meaningful *value shifts* pertained to the importance of essential constituents of health and wellbeing. These shifts often led to greater gratitude, appreciation for an immediate social environment with an adequate understanding of the disease, and the capacity for compassion with strangers suffering from ill health. Moreover, some survivors communicated an intense drive for *self-determination* or independence regarding fundamental life choices and decisions around their medical care associated with having overcome the hardship of critical illness. In the medical domain, the dynamics of self-determination played out in different ways among different family members. While some reported relinquishing the responsibility for medical decisions entirely to trusted professionals, others were unable to accept professional help, even at the cost of their health.

For representative quotes related to the theme outlined above, see Table 6.

## Temporal boundaries

Whereas the period following the appearance of the first blisters leading into systemic decline was perceived as a relentless and overwhelming rush, survivors soon found themselves in the chaos of delirium and sensory deprivation.

**Table 5. Interview data demonstrating themes related to limits of understanding.**

| |
|---|
| **Reduced situational understanding** |
| *DZH004 (survivor): So, I was then transferred to [city where hospital care took place], and, um, it was like that – so first came the first surgery – that was the one where all my skin was taken off and artificial skin was put on – nobody told me that I would come out of there as a mummy... And then I come out, suddenly have a bladder catheter... I'm completely bandaged up everywhere... I'm in really bad pain... That was... That was definitely a shock at first.* |
| **Existential distress** |
| *DZH003a (survivor): So mentally, I don't think I was doing so well at the time. Because they put me on cannabis at some point because I just... somehow... I can't really remember much about that. I only know what my mother told me about how I somehow tried to blame others or something like that, I think. Or how I struggled with my fate.* |
| **Understanding of others** |
| *DZH007 (survivor): Well, what I personally noticed the most through the illness is that you simply get a different perspective on life. I have to deal with death every day in the hospital and in medical school, but it's not as if you're on their side of it yourself. And when you reflect on it... A doctor once told me that my mortality likelihood was 70 percent. That's just something completely different when you're in it yourself and are told that than when you have to convey it to a patient yourself.* |

**Table 6. Interview data demonstrating themes related to social boundaries.**

**Isolation**
*DZH003a (survivor): I just couldn't really identify with the people in my class... So, I had a few friends that I really liked, but they weren't really people about whom I would say, okay, yes, that I truly like being with them and they understand me. It was all so different from... I mean, I was away for a really long time... and I come back and I look completely different.*

**Value shifts**
*DZH002 (survivor): The focus is also set a bit differently. With respect to, I don't know, things that you do at 19 or so, 20, where you say, 'What, that's where you get your self-confidence from?' Now, I would think, what crap is that? (Laughs) doesn't do anything for me at all. The value system is to some extent, I'll say... it's also a question of whether you can call it better, but it's different and perhaps better suited to my situation and depends on other things. Yes, you don't pay as much attention to things that you used to pay attention to before.*

**Self-determination**
*DZH003a (survivor): Yes, I think that aspect [that made me confident again] was really sports and so on, the fact that I can... that I can really have control over my body... can decide for myself what my body looks like... and that I'm proud... I am what I have achieved. That really made a world of difference. I used to… at the outdoor pool, they had these neoprene long-sleeved shirts. And I wore these because I didn't want people to see it. And now, I'm just proud of what I've achieved, and it's actually almost not there anymore. Not at all anymore.*

Eventually, this state of suspension turned into the monotony of slow-paced hospital routines extending into tedious after-care, all of which starkly contrasted with the familiar rhythm of everyday life before the skin reaction. The varied nature of time, along with the survivors' compromised physical and mental fitness, required *patience* from caregivers, relatives, friends, and, most of all, the patients themselves, especially when uncertainty about the expected time course of their hospital stay could not be resolved by the care team.

Despite attempts to reconnect to their previous lives, survivors often experienced a profound *discontinuity* between their pre- and post-EN days. Survivors generally narrated this experience as building a path into a more fulfilled future fitting their true needs. In some, this *forward-looking perspective* played out in seeking deeper connections with loved ones while abandoning superficial relationships, in others as pursuing careers that allowed for flexibility in scheduling work and downtime. To make room for meaningful activities, the pursuit of self-realisation universally involved letting go of external demands either in terms of social expectations or standards of physical beauty but also manifested materially as giving up university programs or, in one case, a long-standing company. For a majority of participants, however, the late stages of recovery were characterised by a better, more compassionate understanding of one's own time needs as well as those of others.

For representative quotes related to the theme outlined above, see Table 7.

## Discussion

In contrast to earlier studies that reported attitudes of mistrust in the medical system associated with the predominantly iatrogenic nature of the disease, [16,17] our results paint a more nuanced picture. Even though many of the participants

**Table 7. Interview data demonstrating the theme of temporal boundaries.**

**Patience**
*DZH007 (survivor): Yes, I think progress has felt slow for a long time. You wait for the dressing change to see what has already changed, and then you're a bit disappointed that it's not that much yet. Maybe you should communicate this more with the patients, that there is no other unexpected organ damage now that you're still in hospital for but that the skin simply needs time to heal and that it's normal to wait weeks.*

**Discontinuity**
*DZH003 (family member): And yes, a child... he really wasn't anymore. So, you could say that as a child he brought heaps of misery home with him – he was really looking forward to seeing his friends, but it just didn't fit anymore. So... Yes, that... they just wanted to play. Of course, he still did, sometimes, when he was ten, but...*

**Forward-looking perspective**
*DZH004 (survivor): In the end, everything turned out well, and now you look towards the future and just change something going forward. Starting with my job, that I'm going to change my schedule and even, I don't know, that I'm going to change my diet. And that I generally review my life goals again, etcetera…*

experienced feelings of powerlessness and dissatisfaction with some of the treatment decisions made, narratives of empowerment and a willingness to navigate the ongoing recovery collaboratively with medical providers were equally prominent. One possible explanation is that most of our participants had received early guidance through the German Registry of Severe Skin Reactions, including extensive advice on safe medical treatment and reassurance about the statistically very low risk of recurrence. For many, the registry remained a consistent point of contact, offering ongoing support as new medical challenges arose.

Despite presenting at different recovery stages, survivors shared a belief in moving towards a state of well-being that was less based on achieving traditional career goals but grounded in a conscious life aligned with one's physical capacity. Based on the participants' responses to the question 'Which factors have supported your recovery the most?", external support across four identified areas, agency, adjustment, normality, and care, played a crucial role in shaping this upward trajectory. These areas present fields of action for healthcare providers who wish not only to contribute their medical expertise to the patient's recovery but also to support them in restoring boundaries essential to a meaningful life (see Table 8).

The goal of this study was to aid guideline developers in envisioning a type of care attentive to underrepresented needs unique to the EN experience. Not all these needs are amenable to medical remediation, however, and limitations as to what can be addressed in guideline recommendations apply.

## Agency

Due to the disorientation ensuing from the overwhelming sensory profile of EN – characterised by severe pain, systemic exhaustion, anaesthesia, or delirium – and an often extremely restricted freedom of movement, loss of agency figures heavily in the illness experience. Aside from suffering physical containment, especially in the early days of their hospital stay, patients can have limited situational awareness and may struggle with making sense of their ongoing medical care [36]. Dependence on caretakers and life-support technology, such as mechanical ventilation, can exacerbate survivors' powerlessness [37,38]. This was also evident in our study, as survivors recounted difficulties with understanding and actively participating in decisions around their care. Healthcare providers should, therefore, strive to actively engage their patients in the recovery process. This could involve discussing and encouraging participation in treatment decisions, including but not limited to analgesia and the provision of sedatives, as well as promoting self-care activities [39,40]. In addition, patients should receive sufficient information on the disease and common sequelae to enable informed decisions and reduce feelings of distress caused by unexpected complications [41,42].

## Adjustment

Throughout recovery, survivors need to adapt to changing bodily demands. In our study, this involved extensive adaptations around self-image, expectations, and behaviour. These adaptations follow a complex psychological and biographical

**Table 8. Fields of action for healthcare providers to facilitate extended recovery and biographical integration.**

| |
|---|
| **Agency** |
| • *Provide physical empowerment* |
| • *Engage in shared information management* |
| **Adjustment** |
| • *Discuss medically required adaptations and accommodate unusual care needs* |
| • *Foster acceptance and appreciation of personal boundaries* |
| **Normality** |
| • *Provide opportunities to connect with lost everyday life* |
| • *Aid in the construction of a new normal* |
| **Care** |
| • *Allow for distributed responsibility between healthcare providers and family members* |
| • *Show empathy and understanding for exceptional impact of the disease.* |

reorientation process and call for flexibility in both personal attitudes and engagement with one's social environment. Where possible, flexibility in terms of openness to deviations from the medical routine can be modeled by healthcare providers, who can discuss and offer available accommodations to meet individual care needs, for example, by granting greater leeway regarding family members' visits to the ICU [43].

Apart from adaptations of a practical nature, patients and family members undergo psychological adaptations that may or may not appear appropriate to the given situation, ranging from denial or repression to dissociation. Implementing screening procedures could help identify individuals likely to benefit from early psychotherapeutic involvement and reduce the risk of psychiatric sequelae, such as post-traumatic stress and anxiety disorders [7,11,44–46].

Regarding the timeframe for psychotherapeutic support for both patients and their family members, our study revealed varying preferences. While some participants who had not received psychotherapy during the acute phase believed they could have benefitted from early consultations, others explicitly dismissed their potential value. Reasons ranged from frustrating experiences with practitioners unfamiliar with the disease over the inability to engage with the topic when struggling for bare survival to a desire to conceptualise their illness as a purely biomedical issue. One participant, who had entered psychotherapy only years into their recovery, praised its unexpected benefits, suggesting an ongoing need for psychological processing at this later stage. The diversity of preferences represented in our study, therefore, does not allow for clear recommendations regarding when to initiate psychotherapy but points to a more tailored approach considering personal capacities and needs.

## Normality

Upon entering the unfamiliar and sometimes alienating space of intensive medical care, study participants commonly experienced a violent rupture with their everyday lives while interactions with the care team focused on immediate needs and treatment decisions to the exclusion of the outside world. As important contacts, medical professionals can engage their patients in conversations on topics situated outside the hospital and help create moments of normality even during this disruptive period.

In their study on patients' responses to the existential threat of cancer, Baker et al. distinguish between coexisting types of normality simultaneously consistent with both continuity and discontinuity with patients' previous lives [47]. While a desire for continuity or 'getting back to normal' will likely be anticipated and supported by medical professionals and family members alike, assisting in constructing a 'new normal' may require different steps. A potentially overlooked avenue for creating normality of this kind is connecting with other survivors of critical illness, who can offer guidance based on their lived experience or, as one medically trained survivor put it, "having been at the other side of the patient's bed".

## Care

Periods of extreme powerlessness and dependence implicate family members of survivors in the care process. These family members, in providing reassurance and emotional support to their loved ones, are often stretched beyond their limits, evidenced by reports of emotional and physical overwhelm recorded in our study.

By its very nature, medical knowledge intersects with the experiential knowledge of the affected only at the surface, leaving essential aspects of what constitutes individualised care under-specified [48–51]. Healthcare providers sensitive to the complementary nature of medical and familial care can better navigate the distribution of responsibilities through open conversations. Aside from discussing individual needs, this could involve outlining potential accommodations such as caregiver leave and mediating access to psychological treatment and social services to ease the burden of family members.

## Limitations

Qualitative research works by way of subjective processing of context-dependent data, whereby it produces rich interpretations closely aligned with participants' own accounts – however, with inherently limited generalisability across broader populations and settings. Despite the validity of strongly representing the researcher's perspective in qualitative inquiry,

given that their reflexivity, theoretical perspective, and interpretive frameworks are vital to uncovering shared meaning across participants' accounts, the absence of triangulation measures other than member checking (i.e., data collection and analysis by a single researcher) may have resulted in misrepresentation and further limited the generalisability of our findings. Additionally, due to the rarity and often fatal disease course of EN, populations at exceptionally high risk, such as the elderly, are absent from this report.

Limitations also apply to the investigation of gender- and age-related differences in the illness experience, which was not undertaken in our study. Even though participants' reports touched on matters of social identity and certain medical categories, such as reproductive health, which could have been read in a gender- or age-sensitive way, we decided not to conduct such an analysis. Primarily, our decision rests on the wish to avoid stereotyping based on a small sample, for example interpreting the differential expression of recovery goals between participants (i.e., recovery of skin appearance vs physical fitness) as necessarily attributable to gender or age. However, our dataset contains indications of gender- and age-specific psychosocial impacts, even if these were only addressed implicitly. Such issues could be explored more systematically in future qualitative and quantitative research. It is worth noting that our sample included a relatively balanced distribution of female (n = 8) and male (n = 6) participants in different age groups (<25: n = 3, 25–59: n = 8, > 60: n = 2) reducing the likelihood of underrepresentation. We hope that future studies building on these preliminary findings can provide a more comprehensive and accurate picture.

## Conclusion

In our study, we have attempted to capture unmet needs, build patient representation, and hopefully improve the management of epidermal necrolysis across care settings.

Medical professionals treating EN should, as much as possible, actively strengthen their patients' agency in processes of shared decision-making as well as foster adjustment to anticipated impairment. They can provide a sense of normality by addressing topics outside of immediate medical care and encouraging connections with other survivors. Finally, they can alleviate at least part of caregivers' distress by acknowledging their burdens and ensuring appropriate support.

In its existential significance, EN shares common features with other critical illnesses that create a violent rupture with past life experiences and necessitate the formation of an identity attuned to bodily needs. It is also markedly different in aspects that take their true shape in the later stages of recovery and are easily overlooked from a perspective concerned with immediate survival rather than the recovery of essential personhood. The experience of skin failure is one of profound destabilisation and reconstitution of boundaries. Removing a basic sense of trust in and acquaintance with one's ailing body, it is also an experience of incomplete understanding – encountered in the self as well as the other. In expanding their perspective beyond the domain of medical facts to encompass the lived experience of long-term sufferers, healthcare providers can tend to their patients not only at the level of the skin but also reach the person underneath.

## Supporting information

**S1 File. Additional information regarding recording and software used and Tables S1-S6.**
(DOCX)

## Author contributions

**Conceptualization:** Ruben Heuer, Maren Paulmann, Alexander Nast.

**Data curation:** Maren Paulmann, Maja Mockenhaupt.

**Formal analysis:** Ruben Heuer.

**Funding acquisition:** Maja Mockenhaupt, Alexander Nast.

**Methodology:** Ruben Heuer.

**Project administration:** Ruben Heuer, Alexander Nast.

**Writing – original draft:** Ruben Heuer.

**Writing – review & editing:** Ruben Heuer, Maren Paulmann, Maja Mockenhaupt, Alexander Nast.

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
