## [Decision Letter · Decision Letter 0]

9 Jul 2025

Dear Dr. Heuer,

Thank you for submitting your manuscript to PLOS ONE. After careful consideration, we feel that it has merit but does not fully meet PLOS ONE’s publication criteria as it currently stands. Therefore, we invite you to submit a revised version of the manuscript that addresses the points raised during the review process.

We look forward to receiving your revised manuscript.

Kind regards,

Keiko Hosohata, Ph.D.

Academic Editor

PLOS ONE

Journal Requirements:

Funding sources (RH, MP, MM, AN): This research was funded through the federal joint committee of the joint self-government of physicians, dentists, hospitals, and health insurance funds in Germany (https://www.g-ba.de/; grant number 01VSF21002).

3. Thank you for stating the following in the Competing Interests/Financial Disclosure section:

Conflict of Interest Disclosures: Maja Mockenhaupt reported serving on a scientific advisory board for Boehringer Ingelheim and Merck AG; the German Registry on severe skin reactions (dZh) received financial support for research (through contracts with the University of Freiburg) from several pharmaceutical companies for the provision of safety information related to drugs manufactured or distributed by them.

Ruben Heuer, Maren Paulmann, and Alexander Nast declare no potential conflicts of interest involving the work under consideration for publication.

We note that one or more of the authors are employed by a commercial company: Boehringer Ingelheim and Merck AG

4. We note you have included a table to which you do not refer in the text of your manuscript. Please ensure that you refer to Tables 2, 3, 4, 5, 6, 7, and 8 in your text; if accepted, production will need this reference to link the reader to the Tables.

Reviewers' comments:

Reviewer's Responses to Questions

**Comments to the Author**

1. Is the manuscript technically sound, and do the data support the conclusions?

Reviewer #1: Yes

Reviewer #2: Yes

2. Has the statistical analysis been performed appropriately and rigorously?

Reviewer #1: I Don't Know

Reviewer #2: Yes

3. Have the authors made all data underlying the findings in their manuscript fully available?

Reviewer #1: Yes

Reviewer #2: No

4. Is the manuscript presented in an intelligible fashion and written in standard English?

Reviewer #1: Yes

Reviewer #2: Yes

Reviewer #1: I like the study you presented, I believe there is an important role to deepening the understanding of the social, personal and emotional impact of SJS/TEN in our patients (also applicable to other diagnoses). However, the manuscript is dense to read and I believe that the information can be delivered in a shorter and less redundant manuscript. I like the figure you provided and that should be added in the paper and could help summarize the words from the paragraphs. Make the paper lighter to read and more engaging because the information is valuable and should be available to the medical community.

Reviewer #2: Comments to authors

General comments

The topic is interesting, and enlightening, and under-recognized in literature. The gained insights would help in better managing patients with epidermal necrolysis. The approach to the topic is sound and the procedure followed is thorough and accurate. Data presentation is comprehensive and logical. The language used is appropriate and correct.

Some questions need to be addressed in general. These include:

- How does experience of EN differ from that of skin burns?

- Why weren`t there comparisons with previous literature on the topic in the discussion section? with emphasis on what knowledge gaps addressed in this research that was lacking in other similar studies.

- Exclusion criteria were not mentioned in the manuscript.

- Sex of the participants also is not mentioned, and whether there were differences in disease perception and outcomes discussed between males and females?

- The impact of age category also on the discussed variables is not discussed.

- I think that a separation of the perspective of caregivers from that of the patients should have been done, rather than collectively presenting their data.

- Effects of oral mucosal affection on the patients that led to feeding restrictions, necessity for tube feeding sometimes, are not addressed by the manuscript.

-

Specific comments

Introduction

- P2, line 3: “alongside pre-reaction or even improved levels” please explain, and write in a clearer way.

**Do you want your identity to be public for this peer review?** For information about this choice, including consent withdrawal, please see our Privacy Policy

Reviewer #1: No

Reviewer #2: No

---

## [Author Response · Author response to Decision Letter 1]

22 Aug 2025

Response to reviewers:

Reviewer #1: I like the study you presented, I believe there is an important role to deepening the understanding of the social, personal and emotional impact of SJS/TEN in our patients (also applicable to other diagnoses). However, the manuscript is dense to read and I believe that the information can be delivered in a shorter and less redundant manuscript. I like the figure you provided and that should be added in the paper and could help summarize the words from the paragraphs. Make the paper lighter to read and more engaging because the information is valuable and should be available to the medical community.

Reply:

Thank you very much for your detailed and thoughtful feedback. As you suggested, we moved Supplementary Figure 1 to the main manuscript. Additionally, we have split up long sentences, removed redundancies, and worked to improve overall readability. In addressing these points, as well as incorporating the suggestions from Reviewer #2, we were unfortunately unable to reduce the overall word count further without compromising the clarity or completeness of the content. We hope, however, that the revisions have enhanced the flow and accessibility of the manuscript, and that its current length is justified by the relevance of the material presented.

Reviewer #2:

General comments

The topic is interesting, and enlightening, and under-recognized in literature. The gained insights would help in better managing patients with epidermal necrolysis. The approach to the topic is sound and the procedure followed is thorough and accurate. Data presentation is comprehensive and logical. The language used is appropriate and correct.

Some questions need to be addressed in general. These include:

1) How does experience of EN differ from that of skin burns?

2) Why weren`t there comparisons with previous literature on the topic in the discussion section? with emphasis on what knowledge gaps addressed in this research that was lacking in other similar studies.

3) Exclusion criteria were not mentioned in the manuscript.

4) Sex of the participants also is not mentioned, and whether there were differences in disease perception and outcomes discussed between males and females?

5) The impact of age category also on the discussed variables is not discussed.

6) I think that a separation of the perspective of caregivers from that of the patients should have been done, rather than collectively presenting their data.

7) Effects of oral mucosal affection on the patients that led to feeding restrictions, necessity for tube feeding sometimes, are not addressed by the manuscript.

Specific comments

Introduction

8) P2, line 3: “alongside pre-reaction or even improved levels” please explain, and write in a clearer way.

Reply:

Thank you for your detailed and valuable feedback. We have addressed your suggestions as follows:

1) We have fleshed out the introductory passage on EN as such:

“In contrast to burn injuries, EN presents as a progressive disease that can leave patients overwhelmed with rapid deterioration [1–3]. Different from burn survivors, many EN-patients also experience extensive mucosal damage, leading to complications of the oral cavity, eyes, and genitals. Among those, conjunctival or corneal epithelial sloughing requiring rigorous ocular surface protection measures, and painful oral erosions impairing oral nutrition are some of the most debilitating [1–3]. Long-term sequelae that demand continued specialist care are also common, including ocular surface scarring, vulvovaginal pain or phimosis [4–8]. Moreover, survivors can suffer deeper psychological consequences of EN, as anxiety around recurrence and avoidance of potentially causative medications can create tremendous emotional strain [4,6–13] .”

2) We have added the following passage in the discussion section:

“In contrast to earlier studies that reported attitudes of mistrust in the medical system associated with the predominantly iatrogenic nature of the disease, [14,15] our results paint a more nuanced picture. Even though many of the participants experienced feelings of powerlessness and dissatisfaction with some of the treatment decisions made, narratives of empowerment and a willingness to navigate the ongoing recovery collaboratively with medical providers were equally prominent. One possible explanation is that most of our participants had received early guidance through the German Registry of Severe Skin Reactions, including extensive advice on safe medical treatment and reassurance about the statistically very low risk of recurrence. For many, the registry remained a consistent point of contact, offering ongoing support as new medical challenges arose.”

3) We have added the following sentences to the methods section:

“We included survivors aged 16 and over with a confirmed diagnosis of EN, and, where applicable, their relatives. We excluded individuals with significant cognitive impairments that could compromise their capacity to provide informed consent.”

4 and 5) We have added the following paragraph to the discussion section:

“Limitations also apply to the investigation of gender- and age-related differences in the illness experience, which was not undertaken in our study. Even though participants’ reports touched on matters of social identity and certain medical categories, such as reproductive health, which could have been read in a gender- or age-sensitive way, we decided not to conduct such an analysis. Primarily, our decision rests on the wish to avoid stereotyping based on a small sample, for example interpreting the differential expression of recovery goals between participants (i.e., recovery of skin appearance vs physical fitness) as necessarily attributable to gender or age. However, our dataset contains indications of gender- and age-specific psychosocial impacts, even if these were only addressed implicitly. Such issues could be explored more systematically in future qualitative and quantitative research. It is worth noting that our sample included a relatively balanced distribution of female (n = 8) and male (n = 6) participants in different age groups (<25: n = 3, 25-59: n = 8, >60: n = 2) reducing the likelihood of underrepresentation. We hope that future studies building on these preliminary findings can provide a more comprehensive and accurate picture.”

Additionally, we have adapted Table 1 to include the participants’ sex as well as to allow for cross-referencing their quotes with the respective demographic information.

6) We appreciate this suggestion. However, the fact that caregivers’ and survivors’ contributions are not treated separately reflects a deliberate decision on our part. There are two main reasons for this. First, we sampled multiple survivors without their caregivers and vice versa. Analysing the data separately would run the risk of juxtaposing experiences that may appear related but are, in fact, independent. Second, the EN-accounts in our study mostly depict a shared illness experience across both groups, with many describing strengthened familial bonds. Where notable differences emerged, for example in the differing timing of feelings of overwhelm between survivors and caregivers in the section Stress Limits/Breaking Points, we highlighted these distinctions explicitly in the text.

7) Except for one participant, who reported persistent sensitivities to acidic foods, none of our interview partners referred to nutritional restrictions during the time of hospitalisation or after. This may either point to a biased sample regarding this particular complication of EN or a feature of the recollection of the early illness experience (e.g., incomplete memory due to sedation, suppression of traumatic or precedence of more impactful memories). We have expanded the introductory section to describe different mucosal manifestations and referenced the requirement of enteral feeding in some patients. We now also briefly discuss the issue food sensitivities in the section Stress Limits/Breaking Points.

8) We have changed the respective sentence to read as follows:

“Accordingly, studies report long-term reductions in quality of life in EN survivors while some maintain or even exceed their self-reported levels of psychosocial wellbeing [4,9–13].”

---

## [Editor Report · Decision Letter 1]

16 Sep 2025

Epidermal necrolysis (Stevens-Johnson syndrome / toxic epidermal necrolysis) as extensive boundary violation: A qualitative study on the illness experience and care needs of survivors in the context of the first German guideline

PONE-D-25-21954R1

Dear Dr. Heuer,

We’re pleased to inform you that your manuscript has been judged scientifically suitable for publication and will be formally accepted for publication once it meets all outstanding technical requirements.

Kind regards,

Keiko Hosohata, Ph.D.

Academic Editor

PLOS ONE
---

## [Editor Report · Acceptance letter]

PONE-D-25-21954R1

PLOS ONE

Dear Dr. Heuer,

I'm pleased to inform you that your manuscript has been deemed suitable for publication in PLOS ONE. Congratulations! Your manuscript is now being handed over to our production team.

Kind regards,

on behalf of

Professor Keiko Hosohata

Academic Editor

PLOS ONE